# Children's spatial language skills predict their verbal number skills: A longitudinal study

Nadja Lindner[1]*, Korbinian Moeller[2,3,4,5], Verena Dresen[6], Silvia Pixner[6], Jan Lonnemann[1,5]

**1** Empirical Childhood Research, University of Potsdam, Potsdam, Germany, **2** Centre for Mathematical Cognition, Loughborough University, Loughborough, United Kingdom, **3** Leibniz-Institut fuer Wissensmedien, Tuebingen, Germany, **4** Department of Psychology, LEAD Graduate School & Research Network, Eberhard Karls University of Tuebingen, Tuebingen, Germany, **5** Center for Research on Individual Development and Adaptive Education of Children at Risk (IDeA), Frankfurt am Main, Germany, **6** Institute of Psychology, UMIT Tirol–Private University for Health Sciences, Medical Informatics and Technology, Hall in Tirol, Austria

* nlindner@uni-potsdam.de

**Data Availability Statement:** We deposit the raw data from the manuscript in an open preservation repository. The link is https://osf.io/n9frw.

## Abstract

The process of number symbolization is assumed to be critically influenced by the acquisition of so-called verbal number skills (e.g., verbally reciting the number chain and naming Arabic numerals). For the acquisition of these verbal number skills, verbal and visuospatial skills are discussed as contributing factors. In this context, children's verbal number skills have been found to be associated with their concurrent spatial language skills such as mastery of verbal descriptions of spatial position (e.g., in front of, behind). In a longitudinal study with three measurement times (T1, T2, T3) at an interval of about 6 months, we evaluated the predictive role of preschool children's (mean age at T1: 3 years and 10 months) spatial language skills for the acquisition of verbal number skills. Children's spatial language skills at T2 significantly predicted their verbal number skills at T3, when controlling for influences of important covariates such as vocabulary knowledge. In addition, further analyses replicated previous results indicating that children's spatial language skills at T2 were associated with their verbal number skills at T2. Exploratory analyses further revealed that children's verbal number skills at T1 predict their spatial language at T2. Results suggests that better spatial language skills at the age of 4 years facilitate the future acquisition of verbal number skills.

## Introduction

Scholastic skills such as mathematical skills are becoming increasingly important to fully exercise citizenship by OECD standards [1]. For instance, Richie and Bates [2] found that mathematical skills of seven-year-olds significantly predicted their later socioeconomic status in adulthood. The acquisition of basic numerical skills thus appears to be relevant for later life prospects. In fact, a meta-analysis by Duncan et al. [3] even indicated that basic numerical skills are more predictive for later academic achievement than reading and attention skills.

According to the developmental model of number acquisition [4], children's numerical development starts with an innate or very early acquired core system for representing

**Funding:** This work was funded by the Deutsche Forschungsgemeinschaft (DFG, German Research Foundation) – 416594961, 491466077. The funders had no role in study design, data collection and analysis, decision to publish, or preparation of the manuscript.

**Competing interests:** The authors have declared that no competing interests exist.

numerical magnitude information (Step 1). This is assumed to be a necessary precondition for children to learn to associate a perceived number of objects or events with spoken (Step 2) and written Arabic symbols (Step 3). Non-symbolic representations of numerical magnitude (step 1), symbolic verbal representations (step 2), and symbolic visual Arabic representations (step 3) are assumed to be integrated by a mental number line representation (step 4), which provides the basis for arithmetic learning. The process of (verbal and visual Arabic) number symbolization (steps 2 and 3) is assumed to be based on the acquisition of so-called verbal number skills, which involve the production of number words (e.g., verbally reciting the number chain and naming Arabic numerals [5]; see also [6]).

For the acquisition of verbal number skills, verbal and visuospatial skills have been discussed as contributing factors [e.g., 7–10]. In this context, Cornu and colleagues [6] assessed vocabulary, phonological awareness, visuospatial skills, verbal and visuospatial working memory as well as verbal number skills of 5- to 6-year-old children and showed that only visuospatial skills emerged as a significant concurrent predictor of verbal number skills. Accordingly, verbal number skills are thought to be spatially grounded [5, 6] and it has been suggested that spatial language skills play a critical role in the acquisition of verbal number skills [5]. Following on from this, the present study pursued the question whether children's spatial language skills predict their verbal number skills longitudinally.

Generally, spatial language can be considered in terms of different categories. For instance, Cannon et al. [11] differentiate between eight categories of spatial language: i) spatial dimensions (e.g., size—big, small); ii) shapes (e.g., square); iii) locations and directions (to describe relative positions); iv) orientations and transformations (e.g., turn right); v) continuous amount (e.g., whole, piece, portion); vi) deictics (e.g., here, there, where); vii) spatial features and properties (e.g., side, curve, round, line); viii) pattern (e.g., next, after, sequence, increase, decrease) [see also 12]. With respect to locations and direction, spatial language has been described as "a means of representing objects and locations through verbal description with respect to multiple [spatial] coordinate systems or frames of reference" [13]. Within this context, it was observed that the majority of four-year-old English-speaking children was able to indicate the position of a teddy placed in (100%), on (90%), under (75%), and in front of a box (75%), while only a smaller part of them was able to indicate the position of the teddy placed behind (50%), above (10%), below (0%), to the left (40%), and to the right (40%) of the box [14]. When asked to place the teddy in the different locations, children's performance was better, but even among seven-year-olds, correct responses of all children were only observed when they were asked to indicate the position of the teddy placed in and on the box, suggesting that children at this age have not yet acquired comprehensive spatial language skills [14]. The present study focused on locative prepositions, this means, spatial language terms belonging to the category "locations and directions". This is based on theoretical considerations according to which the processing of numerical information relies on a spatial representation in the form of a mental number line [e.g., 4, 15], which may unfold in different dimensions (i.e., horizontal, vertical, and sagittal) [see 16] and allow numbers to be spatially localized and determined in their size relation to each other (e.g., 5 comes before 6). In this context, it has been suggested that mastery of spatial language terms might help children to better grasp spatial aspects of numerical representations, such as spatial relations between numbers on a mental number line [5].

Spatial language is considered to play a crucial role in the development of spatial skills [e.g., 17–25] and there are also indications of a potential role of spatial language skills in the development of numerical skills. In this context, Purpura and Reid [26] showed that 3- to 5-year-olds' numerical skills were associated with their so-called mathematical language skills, which were composed of quantitative (e.g., take away, a little bit, more, less, most, and fewest) and

spatial language skills (e.g., nearest, under, first, far, below, in front, middle, end, last, and before). In a similar study, Hornburg and colleagues [27] examined associations between 3- to 6-year-olds' mathematical language and specific basic numerical skills. The authors observed that children's mathematical language proficiency, measured by their quantitative and spatial language skills (comparable to [26]), was significantly associated to children's performance in tasks on verbal counting, one-to-one correspondence, Arabic numeral identification, cardinality understanding, comparisons of sets and/or numerals, ordering numerals, and story problems assessed at the same age. Moreover, a recent training study even indicated that 3- to 5-year-old children who completed a dialogic reading intervention by talking about pre-determined questions that focused on quantitative and spatial language showed significantly improved their numerical skills compared to a business-as-usual control group [28].

Furthermore, in a recent longitudinal study (with a pre- and posttest interval of 6 weeks), Chan et al. [29] investigated associations of 5- to 6-year-old's relational language (combining quantitative, spatial as well as temporal contexts such as more-less, top-bottom, and begin-end) and their number relation skills, assessed using two tasks on cardinal relations between numbers (i.e., number comparison and set relation), one task on ordinal relations (i.e., number ordering), and one task on number-space mapping (i.e., number line estimation). The authors found that relational language skills predicted later number relations skills, particularly for children's performance in number line estimation and even after taking vocabulary, executive functions, but also counting and number identification skills into account.

Instead of examining children's mathematical language skills through a combination of quantitative and spatial language skills, a few studies focused more specifically on spatial language skills of children and their potential association with numerical skills. For instance, Verdine et al. [30] evaluated the relationship between numerical and spatial skills in 3-year-old children and collected parent-reported data on spatial language terms used by parents in interaction with their children (e.g., large, between, below, behind, beside, short, little, on, above, near, in, long, and in front). To assess children's spatial skills, they used a spatial assembly task in which children had to build block constructions from models. Children's numerical skills were assessed in terms of i) counting to the highest number children could do without a mistake, ii) a give-a-number task requiring children to give a specific number of objects (i.e., 3, 1, 2, 4) to the experimenter, iii) naming the successor of given numbers (i.e., 4, 5, 7), and iv) solving nonverbal addition and subtraction tasks using tokens. Results indicated that the number of spatial words parents used in their interaction with children was significantly associated with children's spatial as well as numerical skills. Importantly, these associations remained significant even after controlling for children's general language skills. In addition, Bower, Zimmermann et al. [31] investigated whether training spatial skills of 3-year-old children had an impact on their spatial as well as numerical skills. The training included three different types of feedback strategies for correcting children's mistakes during a spatial assembly task. Children received either corrective feedback, informative gestures or corrective spatial language. In the gesture feedback condition the experimenter followed the shape of i) different shapes (e.g. square, hexagon, rectangles) and ii) puzzle pieces with the fingertip and then asked the child to do the same, while in the spatial language condition the experimenter used spatial words to describe the spatial relation. A transfer effect of spatial training on numerical skills was only found for children with low socioeconomic status who received corrective feedback. In a related study, Bower, Foster et al. [32] showed that 3-year-olds' comprehension of spatial propositions (under, above, between, up, in, on, down, behind, below, middle, in front of, next to, on top of, and upside down) was associated with their spatial and numerical skills, which were assessed concurrently. Moreover, Georges et al. [5] recently showed that 4- to 6-year-old children's spatial language skills assessed by the production and comprehension of spatial

propositions (on, left, before, in, right, behind, above, and under) were related to their verbal number skills even when accounting for the influences of verbal and visuospatial skills, age, sex, and socioeconomic status. According to Georges et al. [5], knowledge of spatial terms might enable children to better grasp spatial aspects of numerical representations, such as spatial relations between numerical magnitudes on a mental number line.

Taken together, there is accumulating evidence for an association between spatial language skills of preschool children and their numerical skills. However, children's numerical skills were often assessed by a composite score including performance on a variety of tasks assessing different basic numerical skills (e.g., magnitude understanding, verbal number skills). In contrast, the findings of Georges et al. [5] suggest that children's spatial language skills are specifically associated with their verbal number skills. To further evaluate this finding, we employed a longitudinal design with three measurement times (T1, T2, T3) and assessed spatial language skills (at T2) of preschool children as well as their verbal number skills (at T1, T2, and T3). Moreover, previous studies predominantly used comprehension tasks with pre-set answer options to assess children's spatial language [32], in which correct answers might occur through guessing or by chance. In fact, children's performance in comprehension tasks was observed to be better than in production tasks [5, 14]. Therefore, we assessed children's spatial language skills using a production task, in which children saw picture cards on which a pet was located relative to an object, and the spatial position of the pet had to be indicated by children. Based on the assumption that spatial language skills play a role in the acquisition of verbal number skills [see 5], we expected that children's spatial language skills (at T2) predict their verbal number skills six month later (at T3). This study also intended to replicate Georges et al.'s [5] finding of an association of children's spatial language skills and their concurrent verbal number skills. Therefore, we ran an analysis assessing the relationship between spatial language skills at T2 and verbal number skills at T2. Considering recent results suggesting an influence of numerical skills on spatial performance [see e.g., 33, 34], we additionally explored whether children's verbal number skills (at T1) might be associated with their spatial language skills 6 month later (at T2). To ensure that potential associations were not driven by individual differences in more general cognitive performance, children's general vocabulary knowledge (assessed at T1), their visual perception skills (assessed at T2), their age (assessed at T1), and sex were included as control variables in multiple linear regression analyses.

## Materials and methods

### Participants

Participants were part of a longitudinal study with 75 German-speaking children comprising four times of measurement. Children's age was assessed at measurement time point 1 (henceforth T1). The average age of the children was 3;10 ($M_{age}$ = 46.31 months, SD 3.07). Further measurement time points (i.e., T2, T3, and T4) followed with an interval of about six months each. In the present study, data of 75 children (41 girls, 34 boys) from T1, T2 and T3 were considered. The reported analysis for the main assumption, whether children's spatial language skills predict their verbal number skills six months later, is based on a sample size of 40 children (including control variables, the analysis is based on a sample size of 39 children), because some participants dropped out due to missing data or as an outlier. Missing data occurred because not all children could be assessed at both measurement time points (T2 and T3, n = 19) or because their protocols were not available for the spatial language skills task and the visual-perception skills task (n = 15, additionally n = 1 for the visual-perception skills task). One child was removed from the sample as an outlier for the spatial language skills task at T2 ($z$ = -2.74). The exploratory analyses were based on the available data.

Importantly, additional analyses indicated no significant differences in mean scores between children who had missing values and those who did not. Additionally, Little's [35] *MCAR* test was calculated for all variables to evaluate whether missing data points were missing at random. The test was not significant (Chi2 = 70.893, df = 61, $p$ = .181) providing no indication for a systematic bias in the distribution of missing data.

Children were recruited from different kindergartens in the state of Tirol, Austria. Written informed consent was obtained from parents or guardians. The study was approved by the local ethics committee *Research Committee for Scientific Ethical Questions* at UMIT Tirol, Hall in Tirol.

## Procedure

Children were tested in one-on-one-sessions in a quiet room in their kindergarten. They completed a test battery with a variety of numerical tasks at the different measurement times, of which only those numerical tasks of T1, T2, and T3 used to measure verbal number skills were considered in this study. At each time point, all tasks were performed in one session. Each session lasted about 30–40 minutes. There was no time limit on the tasks, except for the task to capture children's visual-perception skills. Children's verbal number skills were assessed at T1, T2 and T3, while spatial language skills were only assessed at T2. In addition, children's vocabulary knowledge was assessed at T1 and their visual perception skills at T2. Tasks used were presented in the following order: T1: i) counting, ii) vocabulary part 1, iii) Arabic numerals, iv) vocabulary part 2; T2: i) counting, ii) visual-perception task, iii) spatial language task, iv) Arabic numerals; T3: i) counting, ii) Arabic numerals. More details on the respective measures are provided below.

**Spatial language skills.**   To evaluate children's spatial language skills, they were shown 7 picture cards on which a dog or a cat stood either *in*, *under*, *behind*, *between*, *on*, *in front of* or *next to* a house, flowers, a table, a car, or a box. Children were asked, "Where is the dog?" or "Where is the cat?". All children were shown all picture cards in the same order. Children received one point when they named the spatial position correctly. The number of correctly solved items was used as dependent variable in this task. A detailed list of all items, children's responses and its coding can be found in S1 Table. The reliability of the task indicated by Cronbach's alpha was α = .68.

**Verbal number skills.**   Children's verbal number skills were assessed in two ways: they were asked i) to count as far as possible and ii) to name different Arabic numerals.

Ad i) Children were asked to count aloud as far as possible at T1, T2, and T3 (at T1 up to a maximum of 20, at T2 and T3 up to a maximum of 30). Children were asked "Can you count? Show me!". The largest number up to which children counted correctly was recorded.

Ad ii) Children had to name different Arabic numerals that were presented on cards. At T1 children were asked to name numerals 0, 1, 2, 3, 4, and 5, at T2 they had to name numerals 0, 1, 2, 3, 4, 5, 6, and 7, and at T3 children should name numerals 0, 1, 2, 3, 4, 5, 6, 7, 8, 9, and 10. Arabic numerals were presented to children in random order, ensuring that they were not presented as an ascending sequence. Children received one point for each correctly named Arabic numeral.

The two tasks were significantly correlated at each time point [T1: $r(66)$ = .539; T2: $r(59)$ = .494; T3: $r(58)$ = .493, all p < .001] and combined into one total score for further analysis, following the procedure by Georges et al. [5]. Children's accuracy in both tasks was normed to 1 and the combined score was calculated by summing children's normed scores for both numerical tasks and dividing it by two. The highest sore children could receive on the verbal number skills scale was 1. Retest reliability of the verbal number skills score was indicated by spearman

rank correlations between T1 and T2, $r(52) = .785$ and between T2 and T3, $r(49) = .859$. In this case, Cronbach's alpha was not calculated because the variable counting skills consists of only one item.

**Vocabulary knowledge.** To assess children's general language skills we used the general vocabulary knowledge test *Aktiver Wortschatztest für 3- bis 5-jährige Kinder–Revision* [36]. The test includes pictures reflecting 51 nouns and 24 verbs. Children were asked to say what is shown on each picture or what is done there. The total of 75 items were tested in two shares at T1. For each correctly solved item, one point was awarded. The number of correctly solved items was used as dependent variable. Reliability of the vocabulary task as indicated by Cronbach's alpha was α = .83.

**Visual-perception skills.** To assess children's visual-spatial skills, the *visual-perception* subtest of the *Beery-Buktenica Developmental Test of Visual-Motor Integration* [37] was used at T2. This task focuses on the visual discrimination component and not on motor skills. It comprises a total of 18 items of increasing difficulty. Children had three minutes to complete the tasks. For each item, children had to mark the one out of 2, 3, 4 or 5 geometric shapes presented in a response box below the actual item that fitted the one shape shown as the actual stimulus. For each correctly solved item children received one point. The number of correctly solved items was used as an estimate of children's visual perception skills. The reliability of the task assessing visual perception skills was estimated by Spearman's rank correlations dividing all of the 18 Items in two parts following the odd-even method for split-half reliability: $r(47) = .329$.

## Analyses

Raw data is uploaded at https://osf.io/n9frw.

In the following, we first report correlation analyses measuring bivariate relationships among the different variables. Due to the fact that some variables were not normally distributed we calculated Spearman correlation coefficients (similar results were obtained when the Pearson correlation coefficient was calculated). Multiple linear regression analyses are then reported measuring the effect of spatial language skills at T2 on verbal number skills at T3. This is followed by analysis replicating those of Georges et al. [5], and an additional exploratory analysis.

## Results

Descriptive statistics for all observed variables are shown in Table 1. For the spatial language task, as the task of primary interest, the mean number of items that children solved correctly was 4.53 (SD = 1.5) out of a possible score of 7. The items identified correctly most frequently were "in" (93.9%), "under" (93.9%), "on" (87.8%), and "behind" (79.6%). The items "next to"

**Table 1. Descriptive statistics for the observed variables.**

| Variable | n | M | SD | Theoretical range | Empirical range |
|---|---|---|---|---|---|
| T1: Verbal number skills | 68 | 0.41 | 0.32 | 0–1 | 0–1 |
| T1: Age (in month) | 72 | 46.31 | 3.07 | - | 39–54 |
| T1: Vocabulary knowledge | 72 | 37.43 | 9.78 | 0–75 | 20–61 |
| T2: Spatial language skills | 49 | 4.53 | 1.5 | 0–7 | 2–7 |
| T2: Verbal number skills | 61 | 0.54 | 0.28 | 0–1 | 0.05–1 |
| T2: Visual-perception skills | 49 | 10.76 | 2.45 | 0–18 | 5–16 |
| T3: Verbal number skills | 60 | 0.61 | 0.27 | 0–1 | 0.08–1 |

(44.9%), "between" (30.6%), and "in front of" (22.4%) were performed less well. All seven items were answered correctly by 8 children (16.3%).

## Correlation analyses evaluating uncontrolled bivariate associations between variables

The matrix of all bivariate pairwise correlations is provided in Table 2. Children's spatial language skills at T2 were found to be significantly associated with their verbal number skills at T1, T2, and T3 (see Table 2 and Fig 1). These three correlations, which are the focus of this study, remain significant after controlling for multiple testing using the procedure suggested by Holm [38]. With regard to the control variables employed, vocabulary knowledge was found to be significantly associated with children's verbal number skills at all three measurement times and age at T2 and T3. Visual-perception skills and sex were not significantly associated with either children's verbal number skills or their spatial language skills.

## Multiple linear regression analysis evaluating influences of spatial language skills at T2 on verbal number skills at T3

A multiple linear regression analysis was conducted on children's verbal number skills at T3 to evaluate the relevance of children's spatial language at T2 (see Table 3). The control variables sex, age, vocabulary knowledge, and visual-perception skills were included in the regression model.

Multicollinearity was assessed by examining the variance inflation factor (VIF). The VIF was always below 2, thereby indicating no serious problems of multicollinearity (see Table 3). Residuals were normally distributed.

The results supported our main assumption: Children's spatial language skills at T2 explained a significant amount of unique variance in verbal number skills at T3 after inclusion of the control variables ($\beta$ = .310, $t$ = 2.041, $p$ = .049). None of the other control variables was significantly associated with verbal number skills at T3.

Following the analyses from Georges and colleagues [5], additionally, hierarchical multiple linear regression analysis on children's verbal number skills at T3 was conducted. Results revealed that even though vocabulary knowledge significantly predicted verbal number skills at T3 beyond the influences of the control variables considered in Step 1 ($\beta$ = .420, $t$ = 2.528, $p$ = .016), it was no longer a significant predictor of verbal number skills at T3 after the inclusion

**Table 2. Bivariate pairwise spearman correlation coefficients for the observed variables.**

|   | Variable | 1 | 2 | 3 | 4 | 5 | 6 | 7 |
|---|---|---|---|---|---|---|---|---|
| 1 | T1: Verbal number skills | - | | | | | | |
| 2 | T2: Spatial language skills | .461** | - | | | | | |
| 3 | T2: Verbal number skills | .785*** | .474** | - | | | | |
| 4 | T3: Verbal number skills | .755*** | .352* | .859*** | - | | | |
| 5 | T1: Age (in month) | .227 | -.241 | .315* | .286* | - | | |
| 6 | T1: Sex | .069 | .039 | .024 | .029 | .090 | - | |
| 7 | T1: Vocabulary knowledge | .418*** | .233 | .401** | .462*** | .208 | .048 | - |
| 8 | T2: Visual-perception skills | .255 | -.124 | .232 | .247 | .506*** | .243 | .310* |

*p < .05.

**p < .01.

*** p < .001

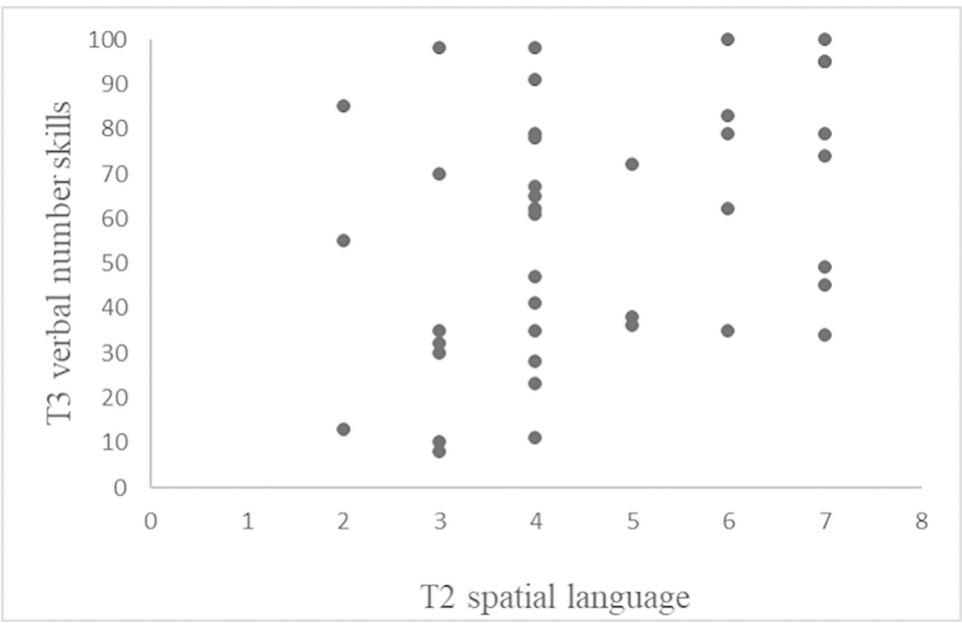

**Fig 1. Correlation between children's spatial language skills measured at T2 and their verbal number skills measured at T3.**

of spatial language skills in the regression model in Step 2 ($\beta$ = .317, $t$ = 1.902, $p$ = .066). None of the other control variables was significantly associated with verbal number skills at T3.

## Multiple linear regression analysis evaluating influences of spatial language skills at T2 on verbal number skills at T2

Next, we conducted another multiple linear regression analysis on children's verbal number skills at T2 to evaluate the relevance of children's concurrent spatial language skills (see Table 4). Again, the control variables sex, age, vocabulary knowledge, and visual-perception skills were incorporated into the regression model.

Multicollinearity was assessed by examining the variance inflation factor (VIF). The VIF was always below 2, thereby indicating no serious problems of multicollinearity (see Table 4). Residuals were normally distributed.

**Table 3. Multiple linear regression analysis on children's verbal number skills at T3.**

|  | VIF | $\beta$ | $R^2$ |
|---|---|---|---|
| **Model** |  |  | .331* |
| Age | 1.526 | .157 |  |
| Sex | 1.188 | .190 |  |
| Vocabulary knowledge | 1.373 | .317 |  |
| Visual-perception skills | 1.566 | -.039 |  |
| Spatial language skills | 1.137 | .310* |  |

VIF, variance inflation factor.

*p < .05.

**p < .01.

*** p < .001.

**Table 4. Multiple linear regression analysis on children's verbal number skills at T2.**

|  | VIF | β | $R^2$ |
|---|---|---|---|
| **Model** |  |  | .442*** |
| Age | 1.482 | .111 |  |
| Sex | 1.127 | .127 |  |
| Vocabulary knowledge | 1.246 | .272 |  |
| Visual-perception skills | 1.466 | .096 |  |
| Spatial language skills | 1.105 | .481*** |  |

VIF, variance inflation factor.

*p < .05.

**p < .01.

*** p < .001.

Consistent with previous findings [see 5], children's spatial language skills at T2 explained a significant amount of unique variance in concurrent verbal number skills after inclusion of the control variables (β = .481, t = 3.675, p < .001). None of the other control variables was found to be significantly associated with verbal number skills at T2.

Again, hierarchical multiple linear regression analysis on children's verbal number skills at T2 revealed that even though vocabulary knowledge significantly predicted verbal number skills at T2 beyond the influences of the control variables considered in Step 1 (β = .402, t = 2.583, p = .014), it was no longer a significant predictor of verbal number skills at T2 after the consideration of spatial language skills to the final regression model in Step 2 (β = .272, t = 1.960, p = .058). None of the other control variables was found to be significantly associated with verbal number skills at T2.

## Multiple linear regression analysis evaluating the effect of verbal number skills at T1 on spatial language skills at T2

Finally, we conducted an exploratory multiple linear regression analysis predicting children's spatial language skills at T2 to evaluate the importance of children's verbal number skills six months earlier at T1 (see Table 5). Again, the control variables sex, age, vocabulary knowledge, and visual-perception skills were considered in the regression model.

Multicollinearity was assessed by examining the variance inflation factor (VIF). The VIF was always below 2, thereby indicating no serious problems of multicollinearity (see Table 5). Residuals were normally distributed.

**Table 5. Multiple linear regression analysis on children's spatial language skills at T2.**

|  | VIF | β | $R^2$ |
|---|---|---|---|
| **Model** |  |  | .288* |
| Age | 1.471 | -.306 |  |
| Sex | 1.207 | .035 |  |
| Vocabulary knowledge | 1.539 | .154 |  |
| Visual-perception skills | 1.642 | -.035 |  |
| Verbal number skills (T1) | 1.567 | .431* |  |

VIF, variance inflation factor.

*p < .05.

**p < .01.

*** p < .001.

Children's verbal number skills at T1 explained a significant amount of unique variance in their spatial language skills measured six month later after inclusion of the control variables ($\beta$ = .431, $t$ = 2.448, $p$ = .019). None of the other control variables was found to be significantly associated with children's spatial language skills at T2.

## Discussion

The aim of this longitudinal study was to evaluate whether preschoolers' spatial language skills predict their verbal number skills. In accordance with the assumption that spatial language skills play a significant role in the acquisition of verbal number skills [see 5], our results substantiated a significant relationship between children's spatial language skills (mean age 52 months) and their verbal number skills assessed six months later. This relationship proved significant when controlling for the effects of important covariates, such as general vocabulary knowledge. By demonstrating that children's spatial language skills predicted their future verbal number skills, the present study extends prior work indicating that children's understanding of spatial language is related to concurrently measured verbal number skills [see 5] by showing that spatial language skills seem to facilitate children's acquisition of verbal number skills in their numerical development.

With respect to the four-step developmental model of number acquisition [4], our results suggest that the assumed development of a spatial mental number line representation (step 4) may already begin before and/or during the process of (verbal and visual Arabic) number symbolization (steps 2 and 3) and–as a consequence–the postulated stages may not be clearly separable. Based on the assumption that verbal number skills are spatially grounded [5, 6], one might argue that the development of a spatial mental number line representation accompanies the process of number symbolization. In this vein, Georges and colleagues [5] suggested that spatial language might promote the development of verbal number skills—which they consider a first milestone in the process of number symbolization—by supporting the spatial representation of numbers on a mental number line [5]. Accordingly, in order to identify the abstract symbols of Arabic numerals and to verbalize their sequence, it might be helpful to mentally localize them spatially. Alternatively, spatial language skills and verbal number skills might be associated due to common requirements in understanding symbolic representations. For instance, Gilligan-Lee and colleagues [39] argue that in order to understand symbolic number, Arabic numerals and number words need to be attached with the quantities they represent, and that understanding spatial terms similarly requires attaching verbal labels with spatial concepts (e.g. the word "above" with the spatial concept of above). As such, children who have developed a better understanding of symbolic (linguistic) representations of spatial concepts might also be more likely to acquire a better understanding of symbolic representations of numerical concepts. To contrast these two accounts, the association between verbal number skills and another category of spatial language (e.g., deictics) which is not expected to support a spatial representation of numbers on a mental number line might be worth assessing and comparing to the association of spatial language and verbal number processing in the present study. This may be desirable to pursue in future studies.

In line with previous findings by Georges et al. [5], children's spatial language skills were also found to be significantly associated with their concurrent verbal number skills. Georges and colleagues [5] reported analyses of variance and correlation analyses indicating that individual differences in age, sex, verbal skills and visuospatial skills were significantly associated with children's verbal number skills. This contrasts with the findings of the present study, showing that vocabulary knowledge was significantly associated with children's verbal number skills at all three measurement time points as well as age with children's verbal number skills at

T2 and T3. In contrast, however, visual-perception skills and sex were not significantly associated with children's verbal number skills. Similarly, contrary to the findings of Georges and colleagues [5], children's spatial language skills were not found to be significantly correlated with any of the control variables in the present study. These discrepancies might be related to the fact that most variables–even though termed similarly–were not assessed in exactly the same way. For example, Georges et al. [5] used backward counting to assess children's verbal number skills, which was not the case in the present study. Moreover, the lack of a significant correlation between children's visual-perception skills and their verbal number skills or their spatial language skills might be due to the relatively low reliability of the visual-perception task in the present study. This should be kept in mind when interpreting the results of the current study. Another difference to the study by Georges et al. [5] is that we did not consider children's socioeconomic status as a control variable. Georges et al. [5] observed significant associations between children's socioeconomic status and their verbal number skills as well as their spatial language skills. However, the association of spatial language skills and verbal number skills proved significant even after controlling for children's socioeconomic status. It thus appears unlikely that an influence of differences in children's socioeconomic status might have biased the present study critically.

Exploratory analysis indicated that children's verbal number skills assessed at T1 (mean age 46 months) predicted their spatial language skills six month later. This is in line with findings suggesting an influence of numerical skills on spatial performance [see e.g., 33, 34]. Similar to the present study, Geer et al. [34] observed both an influence of spatial skills on mathematical skills as well as an influence of mathematical skills on spatial skills in first- through third-grade elementary school children. They suggested this to indicate a reciprocal relationship between the development of spatial and numerical skills. They argue that students who perform better in mathematics might be more likely to use spatial reasoning when solving mathematical problems and that they can further practice and develop their spatial skills through their engagement with mathematics as compared to students who rely less on spatial processing when solving mathematical problems. As such, the findings of the present study might be interpreted in a similar way, as substantiating the idea of a potential reciprocal relationship between preschool children's spatial language skills and their verbal number skills. In this regard, it is interesting to note that our results indicated a stronger relationship between verbal number skills at T1 and spatial language at T2 than that between spatial language at T2 and verbal number skills at T3. However, as our findings are partly based on exploratory analyses, conclusions based on these findings should be taken with caution. Future studies are needed to better understand the potentially reciprocal relation between children's spatial (language) skills and their verbal number skills during their numerical development.

An important point to consider relates to how children's spatial language skills can be assessed. While we assessed spatial language skills using a production task, in previous studies comprehensions tasks [see 27, 32] or both production and comprehension tasks [see 5] were used. The fact that we only assessed spatial language production may be one reason for the relatively low internal consistency of the spatial language task in this study. Georges et al. [5] reported similarly low internal consistency when only considering spatial language production. However, after combining both task variants, internal consistency improved. The relatively low internal consistency of the spatial language task in this study, should be considered when interpreting the present results. However, as we observed results consistent with those of Georges et al. [5], this corroborates the validity of our results.

Another point to consider when measuring spatial language skills is that a comprehensive understanding of verbal descriptions of spatial position involves the consideration of different spatial frames of reference [see e.g., 40]. That is, children need to understand that spatial

relationships can be described from the viewer's perspective and from the perspective of a directed ground entity. By using a non-directed ground entity (a bucket), Bower, Foster and colleagues [32] restricted their spatial language comprehension task to the viewer's perspective. This also seemed to be the case in the study by Georges and colleagues [5], although different ground entities were used and not all were reported (e.g., box, tree, and table). In the present study, however, we used not only non-directed but also directed ground entities (e.g., a car with a clearly identifiable front side) which allowed to describe spatial relationships from the viewer's perspective as well as from the perspective of the directed ground entity. With one exception, in all stimuli there was no conflict between these two perspectives, so that there was a definite correct answer. In one case, the position of a cat relative to a car should be described, with the front of the car pointing to the left from the viewer's perspective. From the child's perspective, the cat was in front of the car (which was considered the correct answer), but from the perspective of the directed ground entity (the car) it was next to the car. Importantly, excluding this stimulus when analyzing the data did not change the results. As such, it would be desirable to evaluate children's consideration of different spatial reference frames more explicitly in future studies on the relationship between spatial language skills and numerical skills.

When interpreting the results of the current study it needs to be considered that the sample size of the study is relatively small. Additionally, it has to be noted, that the assessment of spatial language skills was limited to prepositions from the category "locations and directions" of spatial language (according to [11]). Additionally, the number of items was limited and the task was only administered at, and a single time point. In hindsight, and knowing the results of Bower, Foster et al. [32] who observed an association between spatial language and numerical skills already in 3-year-old children, it would have been desirable having had assessed children's spatial language skills already at T1. In future studies, spatial language and numerical skills should be measured at different time points to be able to investigate relations in different age groups and possible reciprocal relationships more closely.

In conclusion, the present study is the first to show that children's spatial language skills predict their future verbal number skills. Further research is needed to better understand the mechanisms underlying this association. Considering that the acquisition of basic numerical skills in kindergarten is important for children's future academic achievement [3], the results of this study are highly relevant by indicating that fostering children's understanding of specific verbal descriptions of spatial positions, for example in the context of parent-child or educator-child interactions, may positively affect children's future numerical development in the long run. As such, fostering children's spatial language skills may be a successful way to stimulate numerical development even before formal mathematics instruction begins [5], thereby also encouraging early STEM education.

## Supporting information

**S1 Table. Sequence of items, children's responses translated from German into English, and children's responses scored as correct in each case for the spatial language skills task.** (PDF)

## Acknowledgments

We thank the children and kindergartens for their participation and the Tyrolean authorities for approving the study.

## Author Contributions

**Conceptualization:** Verena Dresen, Silvia Pixner.

**Formal analysis:** Nadja Lindner, Korbinian Moeller, Jan Lonnemann.

**Investigation:** Verena Dresen, Silvia Pixner.

**Writing – original draft:** Nadja Lindner, Jan Lonnemann.

**Writing – review & editing:** Nadja Lindner, Korbinian Moeller, Verena Dresen, Silvia Pixner, Jan Lonnemann.

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
