## [Decision Letter · Decision Letter 0]

27 Jun 2022

PONE-D-22-10808Children’s spatial language skills predict their verbal number skills: A longitudinal studyPLOS ONE

Dear Dr. Lindner,

Thank you for submitting your manuscript to PLOS ONE. I have sent it to two expert reviewers and have now received their comments back. As you will see at the bottom of this email, both reviewers found merit in your manuscript, but identify a number of issues that notably relate to theoretical framing, methods, and analytic strategy.Therefore, although I cannot accept your manuscript in its current form, I would be willing to consider a revised version of your manuscript taking into account the reviewers' comments. The revised manuscript would be sent back to the original reviewers. Please submit your revised manuscript by Aug 11 2022 11:59PM. If you will need more time than this to complete your revisions, please reply to this message or contact the journal office at plosone@plos.org. Please include the following items when submitting your revised manuscript:A rebuttal letter that responds to each point raised by the academic editor and reviewer(s). You should upload this letter as a separate file labeled 'Response to Reviewers'.A marked-up copy of your manuscript that highlights changes made to the original version. You should upload this as a separate file labeled 'Revised Manuscript with Track Changes'.An unmarked version of your revised paper without tracked changes. You should upload this as a separate file labeled 'Manuscript'.

We look forward to receiving your revised manuscript.

Kind regards,

Jérôme Prado

Academic Editor

PLOS ONE

Journal Requirements:

3. PLOS requires an ORCID iD for the corresponding author in Editorial Manager on papers submitted after December 6th, 2016. Please ensure that you have an ORCID iD and that it is validated in Editorial Manager. To do this, go to ‘Update my Information’ (in the upper left-hand corner of the main menu), and click on the Fetch/Validate link next to the ORCID field. This will take you to the ORCID site and allow you to create a new iD or authenticate a pre-existing iD in Editorial Manager. Please see the following video for instructions on linking an ORCID iD to your Editorial Manager account: https://www.youtube.com/watch?v=_xcclfuvtxQ.

“This work was funded by the Deutsche Forschungsgemeinschaft (DFG, German Research Foundation) – 416594961.”

“This work was funded by the Deutsche Forschungsgemeinschaft (DFG, German Research Foundation) – 416594961.”

“This work was funded by the Deutsche Forschungsgemeinschaft (DFG, German Research Foundation) – 416594961.”

Reviewers' comments:

Reviewer's Responses to Questions

**Comments to the Author**

1. Is the manuscript technically sound, and do the data support the conclusions?

Reviewer #1: Partly

Reviewer #2: Yes

2. Has the statistical analysis been performed appropriately and rigorously? 

Reviewer #1: Yes

Reviewer #2: No

3. Have the authors made all data underlying the findings in their manuscript fully available?

Reviewer #1: Yes

Reviewer #2: Yes

4. Is the manuscript presented in an intelligible fashion and written in standard English?

Reviewer #1: Yes

Reviewer #2: Yes

5. Review Comments to the Author

Reviewer #1: The question addressed in this paper is interesting and, as the authors state, the results are highly relevant by indicating that fostering children’s understanding of specific spatial terms may positively affect their future numerical development in the long run. Moreover, the paper provides a very elaborate introduction with a lot of background information. Nonetheless, I have some comments and questions that I would like the authors to address in a revision. For instance, to my mind, the methods are not very clear when it comes to the description of the sample size (please see specific comments below). In addition, the results seem a bit rushed. It would have been interesting to also see how relations between the control variables and verbal number skills change when including spatial language in the model (even though this was not the main aim of the study). This might have been especially relevant considering the lack of correlation between visual perception skills and verbal number skills. I also believe that the choice of control measures and the absence of their relations with the variables of interest needs to be discussed in greater detail. For instance, the authors could elaborate on the finding that visual abilities, as indexed by visual perception skills, did not relate to verbal number skills as it was the case in Cornu et al. (2018) and Georges et al. (2021). Furthermore, I believe that the conclusion that spatial language predicts verbal number skills 6 months later even after controlling for visual and verbal abilities should be drawn more carefully, considering the lack of correlations between the control variables and verbal number skills or spatial language. Overall, I believe that the study has great potential but that it needs further elaboration with regards to the analyses. More specific comments are provided below.

Methods:

•I am a little confused concerning the sample size. The authors indicate that: “In the present study, data of 75 children (41 girls, 34 boys) from T1, T2 and T3 were considered”. But then they report that the main analysis including control variables is only based on 36 children due to participants dropping out. By “drop out” do the authors mean that those participants did not follow the study through until the end or do they mean that those children were excluded during the analysis? If they were excluded at the analysis stage, it would mean that more than half of the children had to be removed due to missing values or outlier variables. This seems like a lot. Could the authors please elaborate here?

•Regarding Table S1, why was “inside” for the first item not considered as correct?

•Why was the “give a number task” not considered as a measure of verbal number knowledge?

•Why did the authors decide to calculate a composite score for the two verbal number skill measures, even though they stated in the introduction that it might be useful to separately assess the effect of spatial language on different verbal number skills (“It thus remains unclear which numerical skills are specifically associated with spatial language skills of children.”)?

•The reliability of the visual perception tasks seems fairly low. Might this explain the lack of correlation with any of the other variables?

Results:

•Looking at Table 1, why was data from only 49 children considered when computing the spatial language and visual perception measures, while data from 61 children was used for verbal number skills at T2? Does that mean that a lot of children had missing values or were considered as outliers for the spatial measures?

•In general, the result section seems a bit rushed. Sometimes, it is not clear whether control variables have been taken into account when stating correlation coefficients. If I understand correctly, the correlation coefficients reported in the main text (but not Table 2) are accounted for the influences of the control variables.

•So I assume that when the authors state that: “Children’s spatial language skills measured at T2 significantly predicted verbal number skills measured at T3, r(34) = .382, p = .022”, they accounted for the control variables including vocabulary knowledge at T1 and visual perception skills at T2?

•Here, it is also interesting to see that the relation between spatial language at T2 and verbal number skills at T3 (in Table 2: r = 352) does not change when controlling for the covariates (in the main text: r = .382). Do the authors think that one of the reasons for this might be the lack of correlation between visual perception skills and verbal number skills as well as spatial language at T2? Moreover, vocabulary knowledge at T1 was not related to spatial language at T2. How would the authors justify the inclusion of visual perception skills and vocabulary knowledge as control variables in the relation between spatial language and verbal number skills?

•To my mind, it is also surprising that the relation between spatial language at T2 and verbal number skills at T2 (in Table 2: r = 474) seems to be more pronounced when taking into account the control variables (in the main text: r = .554). Does the inclusion of control variables significantly change the relation between spatial language and verbal number skills? Did the authors consider including a regression analysis to answer this question?

•Moreover, a regression analysis could provide some information on how the relation between visual perception skills and verbal number skills changes when adding spatial language as a predictor of verbal number skills to the model? Even though this is not the main aim of the study, it would still be beneficial to some readers to display these results.

•From the exploratory analysis, it is also interesting to see that the correlation between verbal number skills at T1 and spatial language at T2 is stronger than the correlation between spatial language at T2 and verbal number skills at T3. Is this something that might be worth interpreting when discussing the reciprocal relation between verbal number skills and spatial language?

Discussion:

•The authors could elaborate on the finding that visual perception skills did not relate to verbal number skills at any time point (in contrast to Cornu et al., 2018 and Georges et al., 2021) nor spatial language (in contrast to Georges et al., 2021). In that sense, they might also revise their statement: “Thus, our results support the assumed relevance of spatial processing in reciting number words as well as in naming Arabic numerals [e.g., Cornu et al., 2018)”. Moreover, while vocabulary knowledge correlated with verbal number skills, it did not predict spatial language.

•The authors state that: “In contrast to Georges et al. (2021), this association was observed for forward counting but not backward counting when assessing children’s verbal number skills”. If I understood correctly, the authors did not assess backward counting. Moreover, they did not measure relations between spatial language and forward counting but only looked at correlations with a composite score of verbal number skills. Maybe they could rephrase this statement.

•The authors mention the low internal consistency of their spatial language production task and that the validity of their findings can, however, be assured given the similar findings of Georges et al. (2021). In this vein, might the low reliability of the visual perception task explain the lack of correlation with any of the variables of interest?

Reviewer #2: The present longitudinal study examined the relation between preschoolers’ spatial language (locative prepositions) and verbal math skills. Results suggest a positive link between children’s production of locative prepositions and their concurrent and future verbal math skills. The research is timely and adds to the growing literature on the close relations between spatial and math skills. However, the paper lacks a good theoretical discussion (i.e., the mechanisms of this link). The methods section missed many important aspects, and I suggest using regression analyses. Below are my points that hopefully can help in the revision process:

• The abstract is very broad, and I do not think it represents the details of the study. For example, sample size can be added, different results were not mentioned, and time points were missing.

• The authors can also consider a broader definition of spatial language that includes the size or features of objects as well as objects' relations to each other (see, e.g., Pruden et al., 2011). Then, they need to tell why they chose locative prepositions and specify in the beginning that by the spatial language, they mean locative prepositions.

• The authors presented relevant literature well. However, they could have emphasized the theoretical link between spatial language and math skills. In particular, I think the authors can add some theoretical discussion that would respond to the following questions:

oWhy is it the case that talking about space (i.e., prepositions) relevant to math skills? What about other spatial terms? Would there be a specific link between producing spatial relations between objects and math skills? How strong is this relationship? Would age matter?

• On p. 7, they need to tell which variables they controlled in the analysis for the hypothesis on the link between spatial language and verbal math skills.

• The hypothesis related to verbal skills predicting spatial language six months later was not clear, partly because the time points and tasks at each time point were not explained before the methods section. Therefore, I would start the present study section by adding a few sentences on the general design and questions. Then tell the reader specific questions and hypotheses.

• The participants section needs further information. Mean age and SD for T2 and T3 should be added.

• What were the verbs in the general language skills assessment? I am asking because some verbs such as fly, fall, and run (motion verbs) can have spatial properties and could be counted as spatial. So if there are verbs like this, the authors could take them out in their score for general language skills.

• The authors should have conducted regressions to address their research questions. Specifically:

o For the prediction analysis of spatial language at T2 predicting verbal math skills at T3, the authors should have run a regression analysis adding age and general verbal skills as control variables. However, I do not think they should only report a correlational analysis.

o For T1 math to T2 spatial language, I would also run a regression taking T2 spatial language as the outcome variable and T1 math as the predictor. They can again add age, general language skills and visual perception skills to the analyses. Otherwise, the correlations did not really control for the other possible variables on the link. Last concurrent T2 verbal math and spatial language analyses should be conducted.

• Did the authors get any sex differences in their results?

• I read the discussion assuming that the results would remain the same with the proper analyses the authors will conduct, namely regressions. My main point in the discussion is very similar to my points in the introduction. The authors need to move beyond the relations between variables but explain the mechanisms of the link between spatial language and verbal math skills. Why is it the case that understanding and talking about the relations between objects are associated with verbal math skills? What does spatial language add to children's math understanding? Is it because the task is verbal?

• The authors may discuss both theoretical and practical implications (in relation to STEM education) of their results.

• Another limitation is the spatial language task. The task involves only locative prepositions, but as I mentioned above, spatial language involves many other components, including verbs and adjectives. This should be added as a limitation. The authors currently only discussed the production task as a limitation (which I disagree with, as we usually receive a wider range of responses in production). However, one limitation with the spatial preposition task is the number of trials.

6. PLOS authors have the option to publish the peer review history of their article (what does this mean?). If published, this will include your full peer review and any attached files.

Reviewer #1: No

Reviewer #2: No

---

## [Author Response · Author response to Decision Letter 0]

31 Aug 2022

We wish to thank the editor and the two reviewers for their insightful and helpful comments as well as the opportunity to resubmit a revised version of the manuscript. We have taken all comments and suggestions into consideration when revising the manuscript and are confident that this has considerably improved it.

See below for our detailed point-by-point responses to the reviewers' comments and remarks.

Reviewer #1: 

The question addressed in this paper is interesting and, as the authors state, the results are highly relevant by indicating that fostering children’s understanding of specific spatial terms may positively affect their future numerical development in the long run. Moreover, the paper provides a very elaborate introduction with a lot of background information. 

We wish to thank Reviewer 1 for their positive overall evaluation of our manuscript. 

Nonetheless, I have some comments and questions that I would like the authors to address in a revision. For instance, to my mind, the methods are not very clear when it comes to the description of the sample size (please see specific comments below). In addition, the results seem a bit rushed. It would have been interesting to also see how relations between the control variables and verbal number skills change when including spatial language in the model (even though this was not the main aim of the study). This might have been especially relevant considering the lack of correlation between visual perception skills and verbal number skills. I also believe that the choice of control measures and the absence of their relations with the variables of interest needs to be discussed in greater detail. For instance, the authors could elaborate on the finding that visual abilities, as indexed by visual perception skills, did not relate to verbal number skills as it was the case in Cornu et al. (2018) and Georges et al. (2021). Furthermore, I believe that the conclusion that spatial language predicts verbal number skills 6 months later even after controlling for visual and verbal abilities should be drawn more carefully, considering the lack of correlations between the control variables and verbal number skills or spatial language. Overall, I believe that the study has great potential but that it needs further elaboration with regards to the analyses. More specific comments are provided below.

We are grateful to Reviewer 1 for these important comments and remarks. Please find below our detailed point by point responses.

Methods:

•I am a little confused concerning the sample size. The authors indicate that: “In the present study, data of 75 children (41 girls, 34 boys) from T1, T2 and T3 were considered”. But then they report that the main analysis including control variables is only based on 36 children due to participants dropping out. By “drop out” do the authors mean that those participants did not follow the study through until the end or do they mean that those children were excluded during the analysis? If they were excluded at the analysis stage, it would mean that more than half of the children had to be removed due to missing values or outlier variables. This seems like a lot. Could the authors please elaborate here?

RESPONSE:

We are thankful to Reviewer 1 for this important comment indicating that this was not made clear enough in the manuscript. For better comprehensibility, we have added a paragraph in the revised version of the manuscript on page 8 to better explain sample size matters. Please note that the sample for the main analysis including control variables is based on 39 children. The originally reported number of 36 children was an incorrect count: 

“The reported analysis for the main assumption (i.e., whether children’s spatial language skills predict their verbal number skills six months later), is based on a sample size of 40 children (including control variables, the analysis is based on a sample size of 39 children), because some participants dropped out due to missing data or as an outlier. Missing data occurred because not all children could be assessed at both measurement time points (T2 and T3, n = 19) or because their protocols were not available for the spatial language skills task and the visual-perception skills task (n = 15, additionally n = 1 for the visual-perception skills task). One child was removed from the sample as an outlier for the spatial language skills task at T2 (z = -2.74). The exploratory analyses were based on the available data.”

•Regarding Table S1, why was “inside” for the first item not considered as correct?

RESPONSE: 

We wish to thank Reviewer 1 for this remark and wish to apologize for the problem in translation from German to English that seemed to have occurred here. Three children answered with ambiguous colloquial German terms, namely "drin" or "drinnen", which can be translated as “inside” but also as “indoor”. To make this clearer, we changed the term to “inside/indoor” in the revised version of Table S1.

•Why was the “give a number task” not considered as a measure of verbal number knowledge?

RESPONSE: 

We are grateful to Reviewer 1 for carefully reading our manuscript. This is a relic from an earlier version of the manuscript that we have now removed when revising the manuscript. Following the study by Georges et al. (2021), we focused exclusively on verbal number skills, as described in the procedure section on p. 9: “…a test battery with a variety of numerical tasks at the different measurement times, of which only those numerical tasks of T1, T2, and T3 used to measure verbal number skills were considered in this study.” 

•Why did the authors decide to calculate a composite score for the two verbal number skill measures, even though they stated in the introduction that it might be useful to separately assess the effect of spatial language on different verbal number skills (“It thus remains unclear which numerical skills are specifically associated with spatial language skills of children.”)?

RESPONSE:

We are thankful to Reviewer 1 for pointing this out to us, as our statement is indeed misleading. Following the approach of Georges et al. (2021), we consider verbal number skills as a specific category of numerical skills and contrast this with the majority of existing studies on the relationship between children’s spatial language skills and their numerical skills, in which numerical skills were assessed by a composite score including performance on a variety of different numerical tasks. To make this point clearer, we have removed the respective sentence in question from the revised manuscript and replaced it by the following on p.7: 

“However, children’s numerical skills were often assessed by a composite score including performance on a variety of tasks assessing different basic numerical skills (e.g., magnitude understanding, verbal number skills) In contrast, the findings of Georges et al. [5] suggest that children’s spatial language skills are specifically associated with their verbal number skills.”

•The reliability of the visual perception tasks seems fairly low. Might this explain the lack of correlation with any of the other variables?

RESPONSE:

Again, we wish to thank Reviewer 1 for their detailed reading of our manuscript. This is indeed a relevant aspect to consider in the discussion. Accordingly, the observed lack of significant correlations between some control variables and our variables of interest is now discussed in more detail in the revised version of the manuscript (see p. 18). In this context, we now indicate that 

“…the lack of a significant correlation between children's visual-perception skills and their verbal number skills or their spatial language skills might be due to the relatively low reliability of the visual-perception task in the present study”.

Results:

•Looking at Table 1, why was data from only 49 children considered when computing the spatial language and visual perception measures, while data from 61 children was used for verbal number skills at T2? Does that mean that a lot of children had missing values or were considered as outliers for the spatial measures?

RESPONSE:

As already mentioned above, the number of drop outs occurred because not all children could be assessed at each of the two subsequent measurement time points (T2 and T3, n = 19). As regards the spatial language task and the visual-perception task children were missing at T2 (n = 15, and additionally n=1 for the visual-perception task). One child was removed from the sample as an outlier for the spatial language skills task at T2 (z = -2.74).

•In general, the result section seems a bit rushed. Sometimes, it is not clear whether control variables have been taken into account when stating correlation coefficients. If I understand correctly, the correlation coefficients reported in the main text (but not Table 2) are accounted for the influences of the control variables.

•So I assume that when the authors state that: “Children’s spatial language skills measured at T2 significantly predicted verbal number skills measured at T3, r(34) = .382, p = .022”, they accounted for the control variables including vocabulary knowledge at T1 and visual perception skills at T2?

RESPONSE:

We are grateful to Reviewer 2 for this important note. In addition to correlation analyses measuring bivariate relationships among the different variables (see p. 13), we now also report regression models including the control variables (see p. 14). Following the analyses from Georges and colleagues (2021), additionally, hierarchical multiple linear regression analysis on children’s verbal number skills at T3 was conducted. Thereby, we hope to make it clearer when we report uncontrolled bivariate correlations as compared to results of the regression analyses in which influences of control variables were controlled for: 

“Correlation analyses evaluating zero-order bivariate associations between variables

The matrix of all bivariate pairwise correlations is provided in Table 2. Children’s spatial language skills at T2 were found to be significantly associated with their verbal number skills at T1, T2, and T3 (see Table 2 and Fig 1). These three correlations, which are the focus of this study, remain significant after controlling for multiple testing using the procedure suggested by Holm [35]. With regard to the control variables employed, vocabulary knowledge was found to be significantly associated with children’s verbal number skills at all three measurement times and age at T2 and T3. Visual-perception skills and sex were not significantly associated with either children’s verbal number skills or their spatial language skills. 

Table 2. Bivariate pairwise spearman correlation coefficients for the observed variables.

Fig 1. Correlation between children’s spatial language skills measured at T2 and their verbal number skills measured at T3.

Multiple linear regression analysis evaluating influences of spatial language skills at T2 on verbal number skills at T3

A multiple linear regression analysis was conducted on children’s verbal number skills at T3 to evaluate the relevance of children’s spatial language at T2 (see Table 3). The control variables sex, age, vocabulary knowledge, and visual-perception skills were included in the regression model. 

Multicollinearity was assessed by examining the variance inflation factor (VIF). The VIF was always below 2, thereby indicating no serious problems of multicollinearity (see Table 3). Residuals were normally distributed.

The results supported our main assumption: Children’s spatial language skills at T2 explained a significant amount of unique variance in verbal number skills at T3 after inclusion of the control variables (β = .310, t = 2.041, p = .049). None of the other control variables was significantly associated with verbal number skills at T3. 

Following the analyses from Georges and colleagues [5], additionally, hierarchical multiple linear regression analysis on children’s verbal number skills at T3 was conducted. Results revealed that even though vocabulary knowledge significantly predicted verbal number skills at T3 beyond the influences of the control variables considered in Step 1 (β = .420, t = 2.528, p = .016), it was no longer a significant predictor of verbal number skills at T3 after the inclusion of spatial language skills in the regression model in Step 2 (β = .32, t = 1.902, p = .066). None of the other control variables was significantly associated with verbal number skills at T3.

Table 3. Multiple linear regression analysis on children’s verbal number skills at T3.

Multiple linear regression analysis evaluating influences of spatial language skills at T2 on verbal number skills at T2

Next, we conducted another multiple linear regression analysis on children’s verbal number skills at T2 to evaluate the relevance of children’s concurrent spatial language skills (see Table 4). Again, the control variables sex, age, vocabulary knowledge, and visual-perception skills were incorporated into the regression model. 

Multicollinearity was assessed by examining the variance inflation factor (VIF). The VIF was always below 2, thereby indicating no serious problems of multicollinearity (see Table 4). Residuals were normally distributed.

Consistent with previous findings [see 5], children’s spatial language skills at T2 explained a significant amount of unique variance in concurrent verbal number skills after inclusion of the control variables (β = .481, t = 3.675, p < .001). None of the other control variables was found to be significantly associated with verbal number skills at T2.

Again, hierarchical multiple linear regression analysis on children’s verbal number skills at T2 revealed that even though vocabulary knowledge significantly predicted verbal number skills at T2 beyond the influences of the control variables considered in Step 1 (β = .402, t = 2.583, p = .014), it was no longer a significant predictor of verbal number skills at T2 after the consideration of spatial language skills to the final regression model in Step 2 (β = .272, t = 1.960, p = .058). None of the other control variables was found to be significantly associated with verbal number skills at T2.

Table 4. Multiple linear regression analysis on children’s verbal number skills at T2.

Multiple linear regression analysis evaluating the effect of verbal number skills at T1 on spatial language skills at T2

Finally, we conducted an exploratory multiple linear regression analysis predicting children’s spatial language skills at T2 to evaluate the importance of children’s verbal number skills six months earlier at T1 (see Table 5). Again, the control variables sex, age, vocabulary knowledge, and visual-perception skills were considered in the regression model. Multicollinearity was assessed by examining the variance inflation factor (VIF). The VIF was always below 2, thereby indicating no serious problems of multicollinearity (see Table 5). Residuals were normally distributed.

Children’s verbal number skills at T1 explained a significant amount of unique variance in their spatial language skills measured six month later after inclusion of the control variables (β = .431, t = 2.448, p = .019). None of the other control variables was found to be significantly associated with children’s spatial language skills at T2.

Table 5. Multiple linear regression analysis on children’s spatial language skills at T2."

•Here, it is also interesting to see that the relation between spatial language at T2 and verbal number skills at T3 (in Table 2: r = 352) does not change when controlling for the covariates (in the main text: r = .382). Do the authors think that one of the reasons for this might be the lack of correlation between visual perception skills and verbal number skills as well as spatial language at T2? Moreover, vocabulary knowledge at T1 was not related to spatial language at T2. How would the authors justify the inclusion of visual perception skills and vocabulary knowledge as control variables in the relation between spatial language and verbal number skills?

RESPONSE:

We wish to thank Reviewer 1 for this comment. Indeed, correlation analyses evaluating bivariate associations between the variables indicated that for the control variables employed, vocabulary knowledge was significantly associated with children’s verbal number skills at all three measurement times as well as age with children’s verbal number skills at T2 and T3. In contrast, visual-perception skills and sex were not significantly associated with either children’s verbal number skills or their spatial language skills (see p. 13). We agree with Reviewer 1 that this might actually be one of the reasons for why the strength of the association between spatial language skills at T2 and verbal number skills at T3 does not change substantially after accounting for the control variables. 

In the revised version of the manuscript, we did so by means of multiple regression analyses. In these regression analyses, all control variables were included. In light of the study by Georges and colleagues (2021), in which significant correlations between comparable control variables (i.e., visuospatial abilities and verbal abilities) and spatial language skills as well as verbal number skills were found, we think it is important to consider these control variables in the present study as well. The lack of significant correlations between some of the control variables and our variables of interest as well as their influence on the results of the regression analyses, however, is discussed in more detail in the discussion section of the revised version of the manuscript (see p. 18): 

“Georges and colleagues [5] reported analyses of variance and correlation analyses indicating that individual differences in age, sex, verbal skills and visuospatial skills were significantly associated with children's verbal number skills. This contrasts with the findings of the present study, showing that vocabulary knowledge was significantly associated with children's verbal number skills at all three measurement time points as well as age with children’s verbal number skills at T2 and T3. In contrast, however, visual-perception skills and sex were not significantly associated with children's verbal number skills. Similarly, contrary to the findings of Georges and colleagues [5], children's spatial language skills were not found to be significantly correlated with any of the control variables in the present study. These discrepancies might be related to the fact that most variables – even though termed similarly – were not assessed in exactly the same way. For example, Georges et al. [5] used backward counting to assess children’s verbal number skills, which was not the case in the present study. Moreover, the lack of significant correlations between children's visual-perception skills and their verbal number skills or their spatial language skills, might be due to the relatively low reliability estimated for the visual-perception task in the present study. This should be kept in mind when interpreting the results of the current study.“

•To my mind, it is also surprising that the relation between spatial language at T2 and verbal number skills at T2 (in Table 2: r = 474) seems to be more pronounced when taking into account the control variables (in the main text: r = .554). Does the inclusion of control variables significantly change the relation between spatial language and verbal number skills? Did the authors consider including a regression analysis to answer this question?

RESPONSE: 

We are grateful to Reviewer 1 for this comment. Please note that we replaced the bivariate correlation and partial correlation analyses by a bivariate correlation and a multiple regression analysis to evaluate influences of control variables. The results of the correlation and regression analyses reported in the revised version of the manuscript are slightly different but show a similar trend: Correlation analyses evaluating the bivariate associations between the variables of interest revealed a significant correlation between children’s spatial language skills and their verbal number skills at T2 (r = .474; see p. 13). In addition, the results of the regression model including the control variables showed that spatial language skills at T2 explained a significant amount of unique variance in verbal number skills at T2 when controlling for the influence of control variables (β = .481, t = 3.675, p < .001; see p. 15). However, we do not think that the difference in associations warrants any strong conclusion. In case Reviewer 1 has a specific aspect of these associations that they wish to be discussed we are happy to do so.

•Moreover, a regression analysis could provide some information on how the relation between visual perception skills and verbal number skills changes when adding spatial language as a predictor of verbal number skills to the model? Even though this is not the main aim of the study, it would still be beneficial to some readers to display these results.

RESPONSE:

We wish to thank Reviewer 1 for this important suggestion. In response to this comment, we ran regression analyses and reported them in the revised version of the manuscript (see above for details). As described in the results section, correlation analyses evaluating bivariate associations between the different variables showed that children’s visual-perception skills were not significantly associated with children’s verbal number skills (see Table 2, p. 13). Regression analyses on children’s verbal number skills at T2/T3, including the variables sex, age, vocabulary knowledge, visual-perception skills, and spatial language skills, also revealed no significant associations between children’s visual-perception skills and their verbal number skills as well as no change in the association between visual perception skills and verbal number skills after adding spatial language to the model (see p. 14). 

•From the exploratory analysis, it is also interesting to see that the correlation between verbal number skills at T1 and spatial language at T2 is stronger than the correlation between spatial language at T2 and verbal number skills at T3. Is this something that might be worth interpreting when discussing the reciprocal relation between verbal number skills and spatial language?

RESPONSE:

We are grateful to Reviewer 1 for this interesting comment. We agree that this might be an interesting point to mention when discussing the reciprocal relation between verbal number skills and spatial language. Accordingly, we added a corresponding paragraph to the discussion on p. 19: 

“As such, the findings of the present study might be interpreted in a similar way, as substantiating the idea of a potential reciprocal relationship between preschool children’s spatial language skills and their verbal number skills. In this regard, it is interesting to note that our results indicated a stronger relationship between verbal number skills at T1 and spatial language at T2 than that between spatial language at T2 and verbal number skills at T3. However, as our findings are partly based on exploratory analyses, conclusions based on these findings should be taken with caution. Future studies are needed to better understand the potentially reciprocal relation between children’s spatial (language) skills and their verbal number skills during their numerical development.”

To have a closer look at this, a comparison of listwise, bivariate correlation coefficients of the correlation between children’s spatial language skills and their verbal number skills at T1 versus at T3 was performed using Steiger’s test (Steiger, 1980) implemented in cocor, an R package used for comparison two correlation coefficients (Diedenhofen and Musch, 2015). The comparison of the two correlation coefficients of r = 0.491, and r = 0.409, indicated no significant difference, Z = 1.0485, p = 0.2944. Therefore, we did not discuss this difference in more detail.

Discussion:

•The authors could elaborate on the finding that visual perception skills did not relate to verbal number skills at any time point (in contrast to Cornu et al., 2018 and Georges et al., 2021) nor spatial language (in contrast to Georges et al., 2021). In that sense, they might also revise their statement: “Thus, our results support the assumed relevance of spatial processing in reciting number words as well as in naming Arabic numerals [e.g., Cornu et al., 2018)”. Moreover, while vocabulary knowledge correlated with verbal number skills, it did not predict spatial language.

RESPONSE:

We are grateful to Reviewer 1 for this comment. The lack of significant correlations between some control variables and our variables of interest was also mentioned by Reviewer 1 in an earlier comment. In accordance with the suggestions of Reviewer 1, this aspect is now discussed in more detail in the discussion section of the revised version of the manuscript (see p. 18): 

“Georges and colleagues [5] reported analyses of variance and correlation analyses indicating that individual differences in age, sex, verbal skills and visuospatial skills were significantly associated with children’s verbal number skills. This contrasts with the findings of the present study, showing that vocabulary knowledge was significantly associated with children’s verbal number skills at all three measurement time points as well as age with children’s verbal number skills at T2 and T3. In contrast, however, visual-perception skills and sex were not significantly associated with children's verbal number skills. Similarly, contrary to the findings of Georges and colleagues [5], children’s spatial language skills were not found to be significantly correlated with any of the control variables in the present study. These discrepancies might be related to the fact that most variables variables – even though termed similarly – were not assessed in exactly the same way. For example, Georges et al. [5] used backward counting to assess children’s verbal number skills, which was not the case in the present study. Moreover, the lack of a significant correlation between children’s visual-perception skills and their verbal number skills or their spatial language skills might be due to the relatively low reliability of the visual-perception task in the present study. This should be kept in mind when interpreting the results of the current study.” 

Moreover, to avoid confusion, we have deleted the following sentence from the revised version of the manuscript: “Thus, our results support the assumed relevance of spatial processing in reciting number words as well as in naming Arabic numerals [e.g., 6] and suggest that an understanding of spatial language is facilitative for the development of these verbal number skills.”

•The authors state that: “In contrast to Georges et al. (2021), this association was observed for forward counting but not backward counting when assessing children’s verbal number skills”. If I understood correctly, the authors did not assess backward counting. Moreover, they did not measure relations between spatial language and forward counting but only looked at correlations with a composite score of verbal number skills. Maybe they could rephrase this statement.

RESPONSE:

We are thankful to Reviewer 1 for their detailed reading of the manuscript. In the revised version of the manuscript, we revised the respective paragraph. We now merely note that we did not assess verbal number skills in exactly the same way as Georges and colleagues (2021): 

“For example, Georges et al. [5] used backward counting to assess children’s verbal number skills, which was not the case in the present study” (see p. 18).

•The authors mention the low internal consistency of their spatial language production task and that the validity of their findings can, however, be assured given the similar findings of Georges et al. (2021). In this vein, might the low reliability of the visual perception task explain the lack of correlation with any of the variables of interest?

RESPONSE:

We wish to thank Reviewer 1 for this important point raised. In the course of discussing the lack of significant correlations between some control variables and our variables of interest, we now also point out that 

“…the lack of a significant correlation between children's visual-perception skills and their verbal number skills or their spatial language skills might be due to the relatively low reliability of the visual-perception task in the present study” (see p. 18).

Reviewer #2: 

The present longitudinal study examined the relation between preschoolers’ spatial language (locative prepositions) and verbal math skills. Results suggest a positive link between children’s production of locative prepositions and their concurrent and future verbal math skills. The research is timely and adds to the growing literature on the close relations between spatial and math skills. However, the paper lacks a good theoretical discussion (i.e., the mechanisms of this link). The methods section missed many important aspects, and I suggest using regression analyses. Below are my points that hopefully can help in the revision process:

• The abstract is very broad, and I do not think it represents the details of the study. For example, sample size can be added, different results were not mentioned, and time points were missing.

RESPONSE:

We are grateful to Reviewer 2 for bringing this to pur attention. Accordingly, we revised the abstract as suggested (see p. 2): 

“The process of number symbolization is assumed to be critically influenced by the acquisition of so-called verbal number skills (e.g., verbally reciting the number chain and naming Arabic numerals). For the acquisition of these verbal number skills, verbal and visuospatial skills are discussed as contributing factors. In this context, children’s verbal number skills have been found to be associated with their concurrent spatial language skills such as mastery of verbal descriptions of spatial position (e.g., in front of, behind). In a longitudinal study with three measurement times (T1, T2, T3) at an interval of about 6 months, we evaluated the predictive role of preschool children’s (n = 75, mean age at T1: 3; 10 years) spatial language skills for the acquisition of verbal number skills. Children’s spatial language skills at T2 significantly predicted their verbal number skills at T3, when controlling for influences of important covariates such as vocabulary knowledge. In addition, further analyses replicated previous results indicating that children’s spatial language skills at T2 were associated with their verbal number skills at T2. Exploratory analyses further revealed that children’s verbal number skills at T1 predict their spatial language skills at T2. Results suggests that better spatial language skills at the age of 4 years facilitate the future acquisition of verbal number skills.”

• The authors can also consider a broader definition of spatial language that includes the size or features of objects as well as objects' relations to each other (see, e.g., Pruden et al., 2011). Then, they need to tell why they chose locative prepositions and specify in the beginning that by the spatial language, they mean locative prepositions.

RESPONSE:

We are grateful to Reviewer 2 for this remark. As suggested, we added more information about our understanding of spatial language for the current study to the introduction on p. 4: 

“Generally, spatial language can be considered in terms of different categories. For instance, Cannon et al. [11] differentiate between eight categories of spatial language: i) spatial dimensions (e.g., size - big, small); ii) shapes (e.g., square); iii) locations and directions (to describe relative positions); iv) orientations and transformations (e.g., turn right); v) continuous amount (e.g., whole, piece, portion); vi) deictics (e.g., here, there, where); vii) spatial features and properties (e.g., side, curve, round, line); viii) pattern (e.g., next, after, sequence, increase, decrease) [see also 12]. 

With respect to locations and direction, spatial language has been described as “a means of representing objects and locations through verbal description with respect to multiple [spatial] coordinate systems or frames of reference” [13]. Within this context, it was observed that the majority of four-year-old English-speaking children was able to indicate the position of a teddy placed in (100%), on (90%), under (75%), and in front of a box (75%), while only a smaller part of them was able to indicate the position of the teddy placed behind (50%), above (10%), below (0%), to the left (40%), and to the right (40%) of the box [14]. When asked to place the teddy in the different locations, children’s performance was better, but even among seven-year-olds, correct responses of all children were only observed when they were asked to indicate the position of the teddy placed in and on the box, suggesting that children at this age have not yet acquired comprehensive spatial language skills [14]. 

The present study focused on locative prepositions, this means, spatial language terms belonging to the category “locations and directions”. This is based on theoretical considerations according to which the processing of numerical information relies on a spatial representation in the form of a mental number line [e.g., 4, 15], which may unfold in different dimensions (i.e., horizontal, vertical, and sagittal) [see 16] and allow numbers to be spatially localized and determined in their size relation to each other (e.g., 5 comes before 6). In this context, it has been suggested that mastery of spatial language terms might help children to better grasp spatial aspects of numerical representations, such as spatial relations between numbers on a mental number line [5].”

• The authors presented relevant literature well. However, they could have emphasized the theoretical link between spatial language and math skills. In particular, I think the authors can add some theoretical discussion that would respond to the following questions: Why is it the case that talking about space (i.e., prepositions) relevant to math skills? What about other spatial terms? Would there be a specific link between producing spatial relations between objects and math skills? How strong is this relationship? Would age matter?

RESPONSE:

In providing the rationale for our focus on spatial language terms belonging to the category of "locations and directions" (see above), we have emphasized theoretical considerations about the relevance of these specific spatial terms for the development of numerical skills. Accordingly, the mastery of spatial terms describing spatial relations between objects might be relevant for understanding numbers as well as relations between numbers that are assumed to be represented spatially (see p. 4): “The present study focused on locative prepositions, this means, spatial language terms belonging to the category “locations and directions”. This is based on theoretical considerations according to which the processing of numerical information relies on a spatial representation in the form of a mental number line [e.g., 4, 15], which may unfold in different dimensions (i.e., horizontal, vertical, and sagittal) [see 16] and allow numbers to be spatially localized and determined in their size relation to each other (e.g., 5 comes before 6). In this context, it has been suggested that mastery of spatial language terms might help children to better grasp spatial aspects of numerical representations, such as spatial relations between numbers on a mental number line [5].”

Regarding the other questions raised, the following is what we can say, but from our point of view this is not crucial for our study and we would thus prefer not to mention it in the manuscript:

- What about other spatial terms? It can be assumed, for example, that mastery of spatial terms from the categories "shapes" and "spatial features and properties" is associated with children's geometry development. 

- Would age matter? The described theoretical considerations refer to numerical development and accompanying empirical research has been conducted primarily with children aged 3 to 6 years. More precise theoretical assumptions about the course of development regarding the relationship between spatial language and numerical skills are, to our knowledge, not available. However, in a recent study by Gilligan-Lee, Hodgkiss, Thomas, Patel, and Farran (2021), aged-based differences in spatial language skills as well as relations with different mathematics skills have also been found in children from 6 to 10 years. In this regard, it has been suggested that relational language skills (that include spatial language skills) may influence mathematics skills through numerical skills (see e.g., Chan et al., 2022).

- How strong is this relationship? To our knowledge, there is no theoretical postulate on the strength of the relationship and, as the introduction of our article indicates, there are only a few studies that have specifically investigated the relationship between spatial language and numerical skills, and these studies differ in terms of the task/stimuli used to investigate spatial language and numerical skills as well as in terms of the control variables investigated. Thus, from our point of view, based on the existing data, no serious statement can be made about the strength of the relationship.

• On p. 7, they need to tell which variables they controlled in the analysis for the hypothesis on the link between spatial language and verbal math skills.

RESPONSE:

We are grateful for this note. In addition to correlation analyses measuring bivariate relationships among the different variables, we now also report regression models including the control variables sex, age, vocabulary knowledge, and visual-perception skills. This is now indicated on p. 8 of the revised version of the manuscript: “To ensure that potential associations were not driven by individual differences in more general cognitive performance, children’s general vocabulary knowledge (assessed at T1), their visual perception skills (assessed at T2), their age (assessed at T1), and sex were included as control variables in multiple linear regression analyses.”

• The hypothesis related to verbal skills predicting spatial language six months later was not clear, partly because the time points and tasks at each time point were not explained before the methods section. Therefore, I would start the present study section by adding a few sentences on the general design and questions. Then tell the reader specific questions and hypotheses.

RESPONSE:

We thank Reviewer 2 for this advice and added the following information on p. 7/8: 

- “…we employed a longitudinal design with three measurement times (T1, T2, T3) and assessed spatial language skills (at T2) of preschool children as well as their verbal number skills (at T1, T2, and T3)”

- “Based on the assumption that spatial language skills play a role in the acquisition of verbal number skills [see 5], we expected that children’s spatial language skills (at T2) predict their verbal number skills six month later (at T3). This study also intended to replicate Georges et al.’s [5] finding of an association of children’s spatial language skills and their concurrent verbal number skills. Therefore, we ran an analysis assessing the relationship between spatial language skills at T2 and verbal number skills at T2. Considering recent results suggesting an influence of numerical skills on spatial performance [see e.g., 33, 34], we additionally explored whether children’s verbal number skills (at T1) might be associated with their spatial language skills 6 month later (at T2).”

• The participants section needs further information. Mean age and SD for T2 and T3 should be added.

RESPONSE:

The age of the children was recorded only at T1. This is indicated on p. 8: “Children’s age was assessed at measurement time point 1 (henceforth T1). The average age of the children was 3;10 (Mage = 46.31 months, SD 3.07). Further measurement time points (i.e., T2, T3, and T4) followed with an interval of about six months each.”

• What were the verbs in the general language skills assessment? I am asking because some verbs such as fly, fall, and run (motion verbs) can have spatial properties and could be counted as spatial. So if there are verbs like this, the authors could take them out in their score for general language skills.

RESPONSE:

We thank Reviewer 2 for this hint. The following verbs were used in the test: cutting, building, ironing, combing, swimming, smoking, writing, phoning, jumping rope, drumming, refueling, shouting, sweeping, whistling, peeling, throwing, pulling, tearing, wiping, pinching, kicking, breaking, hammering, knitting. 

Indeed, motion verbs are included, although a clear separation into motion and non-motion verbs appears difficult to us. Moreover, it is important to note in the context of our analyses considering vocabulary as a control variable that the exclusion of motion verbs, should, if at all, reduce the influence of general language skills on the association between spatial language skills and verbal number skills. Therefore, inclusion of these words in the general language measure should not drive our results. As a consequence, we would prefer not to exclude any items.

• The authors should have conducted regressions to address their research questions. Specifically:

o For the prediction analysis of spatial language at T2 predicting verbal math skills at T3, the authors should have run a regression analysis adding age and general verbal skills as control variables. However, I do not think they should only report a correlational analysis.

o For T1 math to T2 spatial language, I would also run a regression taking T2 spatial language as the outcome variable and T1 math as the predictor. They can again add age, general language skills and visual perception skills to the analyses. Otherwise, the correlations did not really control for the other possible variables on the link. Last concurrent T2 verbal math and spatial language analyses should be conducted.

RESPONSE: 

We are grateful to Reviewer 2 for this important suggestion. In response, we conducted regression analyses. In the revised version of the manuscript, we report the results of these multiple linear regression analyses that evaluated i) the effect of spatial language skills at T2 on verbal number skills at T3, ii) the relation between spatial language skills at T2 and verbal number skills at T2, and iii) the effect of verbal number skills at T1 on spatial language skills at T2 (see p. 13):

“Correlation analyses evaluating uncontrolled bivariate associations between variables

The matrix of all bivariate pairwise correlations is provided in Table 2. Children’s spatial language skills at T2 were found to be significantly associated with their verbal number skills at T1, T2, and T3 (see Table 2 and Fig 1). These three correlations, which are the focus of this study, remain significant after controlling for multiple testing using the procedure suggested by Holm [38]. With regard to the control variables employed, vocabulary knowledge was found to be significantly associated with children’s verbal number skills at all three measurement times and age at T2 and T3. Visual-perception skills and sex were not significantly associated with either children’s verbal number skills or their spatial language skills. 

Table 2. Bivariate pairwise spearman correlation coefficients for the observed variables.

Fig 1. Correlation between children’s spatial language skills measured at T2 and their verbal number skills measured at T3.

Multiple linear regression analysis evaluating influences of spatial language skills at T2 on verbal number skills at T3

A multiple linear regression analysis was conducted on children’s verbal number skills at T3 to evaluate the relevance of children’s spatial language at T2 (see Table 3). The control variables sex, age, vocabulary knowledge, and visual-perception skills were included in the regression model. 

Multicollinearity was assessed by examining the variance inflation factor (VIF). The VIF was always below 2, thereby indicating no serious problems of multicollinearity (see Table 3). Residuals were normally distributed.

The results supported our main assumption: Children’s spatial language skills at T2 explained a significant amount of unique variance in verbal number skills at T3 after inclusion of the control variables (β = .310, t = 2.041, p = .049). None of the other control variables was significantly associated with verbal number skills at T3. 

Following the analyses from Georges and colleagues [5], additionally, hierarchical multiple linear regression analysis on children’s verbal number skills at T3 was conducted. Results revealed that even though vocabulary knowledge significantly predicted verbal number skills at T3 beyond the influences of the control variables considered in Step 1 (β = .420, t = 2.528, p = .016), it was no longer a significant predictor of verbal number skills at T3 after the inclusion of spatial language skills in the regression model in Step 2 (β = .32, t = 1.902, p = .066). None of the other control variables was significantly associated with verbal number skills at T3.

Table 3. Multiple linear regression analysis on children’s verbal number skills at T3.

Multiple linear regression analysis evaluating influences of spatial language skills at T2 on verbal number skills at T2

Next, we conducted another multiple linear regression analysis on children’s verbal number skills at T2 to evaluate the relevance of children’s concurrent spatial language skills (see Table 4). Again, the control variables sex, age, vocabulary knowledge, and visual-perception skills were incorporated into the regression model. 

Multicollinearity was assessed by examining the variance inflation factor (VIF). The VIF was always below 2, thereby indicating no serious problems of multicollinearity (see Table 4). Residuals were normally distributed.

Consistent with previous findings [see 5], children’s spatial language skills at T2 explained a significant amount of unique variance in concurrent verbal number skills after inclusion of the control variables (β = .481, t = 3.675, p < .001). None of the other control variables was found to be significantly associated with verbal number skills at T2.

Again, hierarchical multiple linear regression analysis on children’s verbal number skills at T2 revealed that even though vocabulary knowledge significantly predicted verbal number skills at T2 beyond the influences of the control variables considered in Step 1 (β = .402, t = 2.583, p = .014), it was no longer a significant predictor of verbal number skills at T2 after the consideration of spatial language skills to the final regression model in Step 2 (β = .272, t = 1.960, p = .058). None of the other control variables was found to be significantly associated with verbal number skills at T2.

Table 4. Multiple linear regression analysis on children’s verbal number skills at T2.

Multiple linear regression analysis evaluating the effect of verbal number skills at T1 on spatial language skills at T2

Finally, we conducted an exploratory multiple linear regression analysis predicting children’s spatial language skills at T2 to evaluate the importance of children’s verbal number skills six months earlier at T1 (see Table 5). Again, the control variables sex, age, vocabulary knowledge, and visual-perception skills were considered in the regression model. Multicollinearity was assessed by examining the variance inflation factor (VIF). The VIF was always below 2, thereby indicating no serious problems of multicollinearity (see Table 5). Residuals were normally distributed.

Children’s verbal number skills at T1 explained a significant amount of unique variance in their spatial language skills measured six month later after inclusion of the control variables (β = .431, t = 2.448, p = .019). None of the other control variables was found to be significantly associated with children’s spatial language skills at T2.

Table 5. Multiple linear regression analysis on children’s spatial language skills at T2."

In these analyses, all control variables recorded were included (i.e., sex, age, vocabulary knowledge, and visual-perception skills) even though correlation analyses measuring bivariate relationships among the different variables showed that not all of the control variables were significantly associated with our variables of interest. Vocabulary knowledge, for instance, was found to be significantly associated with children’s verbal number skills at all three measurement times as well as age with children’s verbal number skills at T2 and T3, whereas visual-perception skills and sex were not significantly associated with either children’s verbal number skills or their spatial language skills (see p. 13). 

In light of the study by Georges and colleagues (2021), in which significant correlations between comparable variables and spatial language skills as well as verbal number skills were found, we think it is important to consider all control variables. 

The lack of significant correlations between some of the control variables and our variables of interest is discussed in more detail in the revised version of the manuscript (see p. 18): “Georges and colleagues [5] reported analyses of variance and correlation analyses indicating that individual differences in age, sex, verbal skills and spatial-perception skills were significantly associated with children’s verbal number skills. This contrasts with the findings of the present study, showing that vocabulary knowledge was significantly associated with children’s verbal number skills at all three measurement time points as well as age with children’s verbal number skills at T2 and T3. In contrast, however, visual-perception skills and sex were not significantly associated with children's verbal number skills. Similarly, contrary to the findings of Georges and colleagues [5], children’s spatial language skills were not found to be significantly correlated with any of the control variables in the present study. These discrepancies might be related to the fact that most variables variables – even though termed similarly – were not assessed in exactly the same way. For example, Georges et al. [5] used backward counting to assess children’s verbal number skills, which was not the case in the present study. Moreover, the lack of a significant correlation between children’s visual-perception skills and their verbal number skills or their spatial language skills might be due to the relatively low reliability of the visual-perception task in the present study. This should be kept in mind when interpreting the results of the current study. “

• Did the authors get any sex differences in their results?

RESPONSE: 

We thank Reviewer 2 for this comment. No evidence of sex differences was found in either the correlation or regression analyses as sex was not correlated significantly with any of the variables of interest, not found to be a significant predictor in any one of the regression analyses..

• I read the discussion assuming that the results would remain the same with the proper analyses the authors will conduct, namely regressions. My main point in the discussion is very similar to my points in the introduction. The authors need to move beyond the relations between variables but explain the mechanisms of the link between spatial language and verbal math skills. Why is it the case that understanding and talking about the relations between objects are associated with verbal math skills? What does spatial language add to children's math understanding? Is it because the task is verbal?

RESPONSE:

We are grateful to Reviewer 2 for this important remark. In the revised version of the manuscript, we have elaborated more in the discussion section on the potential mechanisms of the relationship between spatial language skills and verbal number skills (see p. 17): 

“Georges and colleagues [5] suggested that spatial language might promote the development of verbal number skills - which they consider a first milestone in the process of number symbolization - by supporting the spatial representation of numbers on a mental number line. Accordingly, in order to identify the abstract symbols of Arabic numerals and to verbalize their sequence, it might be helpful to mentally localize them spatially. Alternatively, spatial language skills and verbal number skills might be associated due to common requirements in understanding symbolic representations. For instance, Gilligan-Lee and colleagues [39] argue that in order to understand symbolic number, Arabic numerals and number words need to be attached with the quantities they represent, and that understanding spatial terms similarly requires attaching verbal labels with spatial concepts (e.g., the word “above” with the spatial concept of above). As such, children who have developed a better understanding of symbolic (linguistic) representations of spatial concepts might also be more likely to acquire a better understanding of symbolic representations of numerical concepts. To contrast these two accounts, the association between verbal number skills and another category of spatial language (e.g., deictics) which is not expected to support a spatial representation of numbers on a mental number line might be worth assessing and comparing to the association of spatial language and verbal number processing in the present study. This may be desirable to pursue in future studies.”

• The authors may discuss both theoretical and practical implications (in relation to STEM education) of their results.

RESPONSE:

We wish to thank Reviewer 2 for this comment. In the revised version of the manuscript, we now mention potential practical implications related to STEM education (see p. 21): 

“Considering that the acquisition of basic numerical skills in kindergarten is important for children’s future academic achievement [3], the results of this study are highly relevant by indicating that fostering children’s understanding of specific verbal descriptions of spatial positions, for example in the context of parent-child or educator-child interactions, may positively affect children’s future numerical development in the long run. As such, fostering children’s spatial language skills may be a successful way to stimulate numerical development even before formal mathematics instruction begins [5], thereby also encouraging early STEM education.”

• Another limitation is the spatial language task. The task involves only locative prepositions, but as I mentioned above, spatial language involves many other components, including verbs and adjectives. This should be added as a limitation. The authors currently only discussed the production task as a limitation (which I disagree with, as we usually receive a wider range of responses in production). However, one limitation with the spatial preposition task is the number of trials.

RESPONSE:

We gratefully picked up these comments on potential limitations of the study and considered them in the discussion section (see p. 20): 

“Additionally, it has to be noted, that the assessment of spatial language skills was limited to prepositions from the category “locations and directions” of spatial language (according to [11]). Additionally, the number of items was limited and the task was only administered at a single time point.”

---

## [Decision Letter · Decision Letter 1]

17 Oct 2022

PONE-D-22-10808R1Children’s spatial language skills predict their verbal number skills: A longitudinal studyPLOS ONE

Dear Dr. Lindner,

Thank you for submitting your revised manuscript to PLOS ONE. I have sent it back to the original reviewers and, as you will see, both think that you addressed their comments and that the manuscript is ready for publication. I agree with that assessment. Before proceeding to the formal decision stage, however, I would like to give you the opportunity to address reviewer #1's point about sample size in the abstract. You may either remove the indication of specify the exact sample sizes used in the different analyses. Please submit your revised manuscript by Dec 01 2022 11:59PM. If you will need more time than this to complete your revisions, please reply to this message or contact the journal office at plosone@plos.org. Please include the following items when submitting your revised manuscript:A rebuttal letter that responds to each point raised by the academic editor and reviewer(s). You should upload this letter as a separate file labeled 'Response to Reviewers'.A marked-up copy of your manuscript that highlights changes made to the original version. You should upload this as a separate file labeled 'Revised Manuscript with Track Changes'.An unmarked version of your revised paper without tracked changes. You should upload this as a separate file labeled 'Manuscript'.If applicable, we recommend that you deposit your laboratory protocols in protocols.io to enhance the reproducibility of your results. Protocols.io assigns your protocol its own identifier (DOI) so that it can be cited independently in the future. For instructions see: https://journals.plos.org/plosone/s/submission-guidelines#loc-laboratory-protocols. Additionally, PLOS ONE offers an option for publishing peer-reviewed Lab Protocol articles, which describe protocols hosted on protocols.io. Read more information on sharing protocols at https://plos.org/protocols?utm_medium=editorial-email&utm_source=authorletters&utm_campaign=protocols.

We look forward to receiving your revised manuscript.

Kind regards,

Jérôme Prado

Academic Editor

PLOS ONE

Journal Requirements:

Reviewers' comments:

Reviewer's Responses to Questions

**Comments to the Author**

1. If the authors have adequately addressed your comments raised in a previous round of review and you feel that this manuscript is now acceptable for publication, you may indicate that here to bypass the “Comments to the Author” section, enter your conflict of interest statement in the “Confidential to Editor” section, and submit your "Accept" recommendation.

Reviewer #1: All comments have been addressed

Reviewer #2: All comments have been addressed

2. Is the manuscript technically sound, and do the data support the conclusions?

Reviewer #1: Yes

Reviewer #2: Yes

3. Has the statistical analysis been performed appropriately and rigorously? 

Reviewer #1: Yes

Reviewer #2: Yes

4. Have the authors made all data underlying the findings in their manuscript fully available?

Reviewer #1: Yes

Reviewer #2: Yes

5. Is the manuscript presented in an intelligible fashion and written in standard English?

Reviewer #1: Yes

Reviewer #2: (No Response)

6. Review Comments to the Author

Reviewer #1: The authors have done a fantastic job in responding to all my questions. I appreciate that the authors have now included regression models in the result section. I also like the fact that they have discussed the low reliability of the visual-perception task and elaborated on the findings that some of the covariates did not correlate with verbal number skills and/or spatial language. I only have a small comment regarding the abstract. I would remove the sample size from the abstract as the main analysis regarding the effect of spatial language on verbal number skills was only conducted on 40 children (and not 75). Apart from that I have no further comments or suggestions and I believe that the study is suited for publication in PlosOne.

Reviewer #2: The authors addressed all questions and comments of the reviewers. I believe this is a good contribution to the literature.

7. PLOS authors have the option to publish the peer review history of their article (what does this mean?). If published, this will include your full peer review and any attached files.

Reviewer #1: No

Reviewer #2: **Yes: **Tilbe Göksun

---

## [Author Response · Author response to Decision Letter 1]

18 Oct 2022

We thank the editor and the two reviewers for their positive assessment of our revision. We have taken Reviewer #1 advice into account in revising the manuscript and have removed the sample size from the abstract.

---

## [Editor Report · Decision Letter 2]

19 Oct 2022

Children’s spatial language skills predict their verbal number skills: A longitudinal study

PONE-D-22-10808R2

Dear Dr. Lindner,

We’re pleased to inform you that your manuscript has been judged scientifically suitable for publication and will be formally accepted for publication once it meets all outstanding technical requirements.

Kind regards,

Jérôme Prado

Academic Editor

PLOS ONE
---

## [Editor Report · Acceptance letter]

21 Oct 2022

PONE-D-22-10808R2 

Children’s spatial language skills predict their verbal number skills: A longitudinal study 

Dear Dr. Lindner:

I'm pleased to inform you that your manuscript has been deemed suitable for publication in PLOS ONE. Congratulations! Your manuscript is now with our production department. 

Kind regards, 

on behalf of

Dr. Jérôme Prado 

Academic Editor

PLOS ONE